# Explainable Automated TI-RADS Evaluation of Thyroid Nodules

**DOI:** 10.3390/s23167289

**Published:** 2023-08-21

**Authors:** Alisa Kunapinun, Dittapong Songsaeng, Sittaya Buathong, Matthew N. Dailey, Chadaporn Keatmanee, Mongkol Ekpanyapong

**Affiliations:** 1Harbor Branch Oceanographic Institute, Florida Atlantic University, Fort Pierce, FL 34946, USA; 2Department of Radiology, Faculty of Medicine, Siriraj Hospital, Mahidol University, Bangkok 10400, Thailand; dsongsaeng@gmail.com (D.S.); sittaya.bua@gmail.com (S.B.); 3Information and Communication Technologies, Asian Institute of Technology, Bangkok 12120, Thailand; dailey.matthew@gmail.com; 4Department of Computer Science, Ramkhamhaeng University, Bangkok 10240, Thailand; chadaporn@ru.ac.th; 5Industrial Systems Engineering, Asian Institute of Technology, Bangkok 12120, Thailand; mongkol@ait.asia

**Keywords:** thyroid nodule, TI-RADS, classification, deep learning, Grad-CAM, heatmap, ResNet, DenseNet

## Abstract

A thyroid nodule, a common abnormal growth within the thyroid gland, is often identified through ultrasound imaging of the neck. These growths may be solid- or fluid-filled, and their treatment is influenced by factors such as size and location. The Thyroid Imaging Reporting and Data System (TI-RADS) is a classification method that categorizes thyroid nodules into risk levels based on features such as size, echogenicity, margin, shape, and calcification. It guides clinicians in deciding whether a biopsy or other further evaluation is needed. Machine learning (ML) can complement TI-RADS classification, thereby improving the detection of malignant tumors. When combined with expert rules (TI-RADS) and explanations, ML models may uncover elements that TI-RADS misses, especially when TI-RADS training data are scarce. In this paper, we present an automated system for classifying thyroid nodules according to TI-RADS and assessing malignancy effectively. We use ResNet-101 and DenseNet-201 models to classify thyroid nodules according to TI-RADS and malignancy. By analyzing the models’ last layer using the Grad-CAM algorithm, we demonstrate that these models can identify risk areas and detect nodule features relevant to the TI-RADS score. By integrating Grad-CAM results with feature probability calculations, we provide a precise heat map, visualizing specific features within the nodule and potentially assisting doctors in their assessments. Our experiments show that the utilization of ResNet-101 and DenseNet-201 models, in conjunction with Grad-CAM visualization analysis, improves TI-RADS classification accuracy by up to 10%. This enhancement, achieved through iterative analysis and re-training, underscores the potential of machine learning in advancing thyroid nodule diagnosis, offering a promising direction for further exploration and clinical application.

## 1. Introduction

Thyroid ultrasound images are invaluable in detecting abnormalities, particularly thyroid nodules, which are specific types of thyroid lesions. Large nodules can interfere with the functioning of other organs, such as blood vessels and the trachea, and some may even mutate and become cancerous. When a radiologist detects a thyroid nodule in an ultrasound image, risk stratification must be performed. This analysis often involves evaluating the nodule’s composition, echogenicity, shape, margin, and echogenic foci.

Many ultrasound-based risk stratification systems (RSS) can be employed to gauge the level of malignancy. The American College of Radiology Thyroid Imaging Reporting and Data System (ACR TI-RADS), published in 2017, is one of the most prevalent [1]. The ACR TI-RADS classifies nodules into five levels (TR1–TR5). Higher levels indicate a need for physicians to perform an FNA (fine-needle aspiration) or thyroid biopsy, while medium levels may lead to follow-up recommendations. The TI-RADS classification system helps prevent over-diagnosis, thus reducing wasted time and cost [2]. For some patients, such as children, a thyroid biopsy may be unsuitable for classifying the malignancy of a nodule. In such situations, the TI-RADS can be a more appropriate diagnostic method [3]. Specific follow-up recommendations for smaller nodules also reduce the chances of missing developing cancers [4].

Although TI-RADS scoring is standardized, it must be performed by an imaging specialist to prevent errors. Hence, an automated system for classifying TI-RADS would be extremely useful. AI models have already achieved accuracy rates of over 85% at binary classification (benign vs. malignancy) [5,6], surpassing radiologists’ average competence of around 74% [7]. However, the lack of tangible risk assessment with such models makes direct comparison with the TI-RADS challenging, keeping ACR TI-RADS widely used.

In some studies, models have been used to analyze and interpret the results of TI-RADS, aiding in the diagnosis and differentiation between benign and malignant tumors at TR4 and TR5 levels [4]. We believe that both techniques are valuable, particularly in an automated system, for predicting the malignancy of a nodule, provided the information used for the prediction is transparent.

Automated feature induction methods such as Grad-CAM are often deemed uninformative, resembling what some refer to as a “blazing box”. However, by combining Grad-CAM with expert rules and explanations, researchers may be able to uncover overlooked insights. Visual interpretation of the results of applying a deep learning model to a medical image is critital, as it helps healthcare professionals understand the decision-making process entrenched in the deep learning model, thereby enhancing trust in the outcome.

These challenges underscore the need for new methodologies that can harness the power of AI while maintaining alignment with established medical guidelines. It is within this context that our work introduces the following novel contributions.

For thyroid nodule classification, the integration of expert rules and Grad-CAM explanations can assist radiologists in identifying aspects missed in TI-RADS or areas where training data are lacking. In this paper, we propose a model to classify nodule type and TI-RADS concurrently. Grad-CAM is employed to analyze the thyroid nodule within the deep learning model, providing an explanation of how the deep learning model views and analyzes the data, and whether its interpretation aligns with that of a radiologist.

The novelty of our approach lies in two key technical contributions. First, we employ Grad-CAM visualization, a technique not previously applied in this context, to augment the understanding and interpretation of deep learning models used for thyroid nodule classification. This integration helps provide new insights and builds trust in the model’s decision-making process. Second, we explore the potential of merging AI classification with the established TI-RADS, aiming to identify nodule types more comprehensively. Unlike prevailing methods that predominantly focus on binary classification of benign and malignant nodules, our study bridges the gap between ML and clinical practice by leveraging both techniques, thus paving the way for more nuanced and precise patient care. These innovations collectively represent an advancement in the field, opening up new avenues for research and clinical practice.

This paper examines the potential for integrating AI classification with the clinician-oriented TI-RADS to identify nodule types. Based on the success of this integration, we plan in future work to explore the use of the AI classification approach towards determination of whether surgery or monitoring should be considered.

## 2. Literature Review

### 2.1. Ultrasound in Diagnosis

Ultrasound is a widely recognized diagnostic tool used in the medical field to detect and confirm various medical conditions, and it plays an essential role in guiding medical procedures and treatments [8].

To create diagnostic images, an ultrasound scan examines the reflection, refraction, scattering, and absorption of sound waves within tissues and organs. The resulting grayscale image provides key insights into the internal structure of the body. Brighter intensities typically represent denser tissues, such as bone and calcium, whereas darker intensities indicate less dense materials, such as blood and water [9].

The thyroid gland, located at a relatively shallow depth, is particularly amenable to ultrasound imaging. Ultrasound Images of the thyroid are often clear and easily interpretable, making ultrasound an invaluable tool for examining this specific gland [10].

### 2.2. Indications for Ultrasound in Thyroid Diseases

Ultrasound examinations are commonly used in the diagnosis and monitoring of thyroid conditions. When a doctor detects a nodule or observes an enlargement of the thyroid gland, ultrasound may be employed to investigate the patient’s condition further [11]. Ultrasound scans are also instrumental in identifying patients at heightened risk of thyroid cancer. This includes individuals with specific conditions such as Hashimoto’s thyroiditis or lymphoma, those who have undergone radiation therapy, or patients experiencing thyroid hormone dysfunction. Furthermore, ultrasound examinations are essential for tracking disease progression or facilitating fine-needle aspiration (FNA). FNA is a critical procedure in which a cell sample is extracted from the targeted nodule for pathological analysis.

### 2.3. Thyroid Nodule and ACR TI-RADS

Thyroid nodules, which can alter the shape of the thyroid gland and potentially impinge on or disrupt the function of neighboring organs such as the trachea, represent one of the most common disease symptoms within the thyroid gland [12]. The risk of malignancy in these nodules is often classified based on characteristics such as shape, composition, calcium content, and fluid content. Intriguingly, some nodules may contain both benign and malignant material at different positions [11].

To assess the nature of thyroid nodules, medical professionals commonly utilize the American College of Radiology Thyroid Imaging Reporting and Data System (ACR TI-RADS) [13]. This system has evolved through multiple revisions, with the latest update being published in 2017 (Figure 1). TI-RADS is a valuable tool in classifying nodules, assigning scores that range from TR1 to TR5 based on features such as composition, echogenicity, shape, margins, and echogenic foci.

Most individuals will develop thyroid nodules as they age, although the vast majority of these nodules are benign. Some, however, can evolve into cancerous growths, necessitating vigilant monitoring and, in cases where the TI-RADS score is high, procedures such as Fine-Needle Aspiration (FNA) or a thyroid biopsy.

The TI-RADS scoring system provides a robust framework for distinguishing between benign and malignant cases. A lower score, such as TR1, indicates a benign condition, while a higher score is more suggestive of malignancy. The category TR3 is often considered a threshold for initiating follow-up, depending on the nodule’s size (15 mm or larger) [14].

### 2.4. Deep Learning for Medical Imaging

The advent of deep learning in computer vision has brought significant advancements to the field of medical image processing in recent years. Numerous methodologies have been developed to assist physicians in various tasks, leveraging algorithms capable of tackling problems such as classification, semantic segmentation, object detection, and 2D/3D synthesis [12,14,15,16,17].

Popular classification models such as ResNet [18], InceptionNet [19], and DenseNet [20] have found widespread application. While most applications employ these models for single output tasks, some have explored their use in multiple output tasks. Specifically, in the context of thyroid nodule classification, deep learning models have been instrumental in assessing nodules’ malignancy.

Some studies have even applied models to classify TI-RADS scores [21]. This application, however, is not prevalent, as malignancy classification is relatively straightforward and often lacks detailed explanation. Moreover, the TI-RADS scoring system is more compatible than binary classification with radiologists’ expert rules; it provides them with clear, comprehensive reporting.

### 2.5. Grad-CAM for Medical Imaging

Gradient-weighted Class Activation Mapping (Grad-CAM) [22] is a visualization technique for deep learning computer vision models. It scores the regions of an input image as to how they influence the output for a specific class.

The Grad-CAM algorithm functions by numerically computing the gradient of the neural network’s output corresponding to the target class with respect to an input. These gradients are visualized as weights on the activations of the input layer, forming a coarse localization map highlighting the significant regions within the input image. The resulting map can underline the parts of the input image most instrumental for determination of the output class.

In the realm of medical imaging, Grad-CAM has potential for many applications. It has been leveraged to visualize regions of an input image contributing most strongly to classification and segmentation model output [23]. Such visualizations assist researchers and clinicians in pinpointing potential zones of concern or interest.

In addition to simple visualization, Grad-CAM can be used as a tool for evaluating a model’s performance and unveiling areas ripe for enhancement. Analyzing the localization maps created by Grad-CAM can enable researchers to detect instances where the model might be placing undue emphasis or lacking focus on specific image regions. Such insights can guide adjustments to the model’s architecture and the refinement of the models’ training data, as we shall see in the forthcoming methodology and results.

## 3. Materials and Methods

### 3.1. Dataset Images and Preparation

We collected 1527 ultrasound images of thyroid nodules from three university hospitals for use as our training and testing data. The data consist of 673 benign and 804 malignant nodules, reflecting the complexity and variety of thyroid nodule characteristics. The data were randomly divided into 1328 training images (including 603 benign and 725 malignant) and 149 test images (including 70 benign and 79 malignant). All cases were diagnosed by doctors who were at least 3rd-year fellows in diagnostic radiology and then confirmed through fine needle aspirations by pathologists.

The features of TI-RADS are integral to our study, as certain features such as comet-tail and spongiform are specifically found in benign nodules and are challenging to detect. Conversely, malignant nodules present more pronounced characteristics. This intricate blend of features required a careful selection of both benign and malignant nodules.

The data were randomly divided as summarized in Table 1.

An image of each nodule, detected and segmented by StableSeg GANs [24] in an ultrasound image, was cropped with a 5% margin and resized to 512 × 512, as shown in Figure 2.

### 3.2. The Deep Learning Model

In this study, we selected two prominent deep learning architectures, ResNet50 and DenseNet201, for classifying thyroid nodules. The decision to use these models was motivated by several factors:

1. Pretrained Weights: Both ResNet50 and DenseNet201 come with weights learned through the large ImageNet dataset, allowing for increased model accuracy without requiring training from scratch;

2. Bypass Layers: The unique characteristics of both models lie in the specific structures of their “bypass” layers. ResNet50 utilizes skip connections that allow the input to jump over some layers, effectively shortening the path, as depicted in Figure 3. This assists in capturing the latest layers’ features while mitigating the vanishing gradient problem.

DenseNet201, on the other hand, takes the bypass concept further by connecting each layer to every subsequent layer in a feed-forward manner, as shown in Figure 4. This dense connectivity approach enables the model to focus on every detail from the beginning, allowing for efficient feature reuse and improving gradient flow.

3. Distinct Focus: We believe that the contrasting focus of ResNet50 and DenseNet201 can provide complementary insights. While DenseNet201 emphasizes the details from the start, ResNet50 is more attuned to the latest layers. These differences may lead to varied results, enriching our understanding of thyroid nodule classification. These well-known and effective models, along with their respective bypass strategies, offer promising capabilities for our specific task, justifying their selection for this study.

### 3.3. Input Images and Multiple Output Classification

Ultrasound images are typically grayscale, but they can be combined with other image types as input to a neural network. Multiple tensors of images can be used to convey different information, such as segmenting nodules to indicate their position. In this paper, we provide three tensors of images: an ultrasound image, a sharpened version of the ultrasound image (enhanced using 3 × 3 max-pooling with stride 1 to reveal micro-calcifications more clearly), and a segmented nodule image to concentrate the model on the nodule. The segmentation is implemented by StableSeg GANs [24], as shown in detail in Figure 5. Previous research has utilized ResNet and DenseNet to classify thyroid nodule ultrasound images as benign or malignant. However, no models have been designed for the multiple fine-grained classifications required by TI-RADS. To address this, we modified the ResNet50 and DenseNet201 classification models from a single output layer to multiple outputs, including TI-RADS types such as composition, echogenicity, margin, and echogenic foci. The model can easily calculate the shape of the nodule from the segmented nodule image, eliminating the need for the deep learning model to learn to predict the nodule’s size. Figure 6 shows the overall input and output of the model.

For each classification output, we utilized cross-entropy loss, chosen for its compatibility Grad-CAM, which provides visual evidence for the analysis. This highlights the multifaceted nature of thyroid nodule characterization. The overall loss function (Equation (Equation 1)) combines various types of loss to capture the complexity of malignant/benign differentiation, composition, echogenic foci, echogenicity, and margin characteristics:(1)$$L_{all}=L_{m}+L_{p}+L_{a}+L_{b}+L_{c}+L_{d}+L_{e}+L_{f},$$
where $L_{m}$ is the malignant/benign loss, $L_{p}$ is the composition loss (solid, mix-cystic, cystic, and spongiform), $L_{a}$ is the echogenic foci of comet-tail loss, $L_{b}$ is the echogenic foci of micro-calcification loss, $L_{c}$ is the echogenic foci of macro-calcification loss, $L_{d}$ is the echogenic foci of peripheral-calcification loss, $L_{e}$ is the echogenicity loss (anechoic, very-hypoechoic, hypoechoic, and hyperechoic), and $L_{f}$ is the margin loss (smooth, ill-defined, irregular, and extension).

### 3.4. Grad-CAM Explanation and Implementation

The goal of this research is not only to develop a deep learning model capable of classifying thyroid nodules, but also to make the decision-making process transparent to expert radiologists. To achieve this transparency, we employed Gradient-weighted Class Activation Mapping (Grad-CAM), a visualization technique that highlights the regions of the input image that are most influential in the model’s decision.

Specifically, Grad-CAM works by generating a coarse localization map of the essential areas for each output type. This is done by examining the gradients of the last 2D feature map before the fully connected layer. The resulting visualized image emphasizes the key regions, allowing radiologists to understand which parts of the image led to the model’s decision.

To further refine the visualized image, we multiply it by the probability of the specific type on which the model is concentrating. This operation helps to eliminate distractions from irrelevant areas and emphasizes the critical features that contribute to an accurate diagnosis. The process can be understood as a filtering mechanism that brings the expert’s attention to what truly matters in the analysis.

The equation below formalizes this step, where ${image}_{gradcam_{original}}$ represents the original visualized image generated by Grad-CAM, and $p_{o}$ is the classification probability for the specific type under consideration:(2)$${image}_{gradcam_{modified}}=p_{o}\times {image}_{gradcam_{original}}.$$

Here, ${image}_{gradcam_{modified}}$ represents the modified Grad-CAM image that incorporates the classification probability $p_{o}$, and ${image}_{gradcam_{original}}$ represents the original unmodified version.

This adjustment to the Grad-CAM visualized output ensures that it aligns with the classification results, offering a coherent and intuitive visual explanation for the clinician to examine. By incorporating the classification probability result into the visualization, we create a more informative and tailored visual representation, bridging the gap between deep learning models and expert radiologists’ interpretation.

## 4. Results

After completing initial training and analyzing the outcomes for nodule malignancy classification and TI-RADS classification, we obtained the accuracy for each classification type over the test set, as presented in Table 2.

In the Grad-CAM visualization, we conducted individual analyses for each classification and sought the expertise of expert radiologists (coauthors DS and SB) to interpret the results. Figure 7 and Figure 8 present sample accurate outcomes along with corresponding expert explanations, while Figure 9 displays a sample incorrect outcome and the experts’ interpretations for those cases.

This study seeks to investigate AI-driven analyses that simulate the cognitive process of radiologists, specifically evaluating discrepancies from the foundational principles outlined by the ACR TI-RADS guidelines. The Grad-CAM results underscore that the AI models’ analytical capabilities are largely congruent with the reasoning of radiologists. However, an exception was noted in the AI system’s inability to differentiate between micro-calcifications and macro-calcifications—distinctions typically made based on calcification size.

Moreover, in instances of purely cystic nodules, the AI model exhibited uncertainty about the interpretation of anechoic structures while successfully identifying both cystic and solid components in mixed-cystic cases. When applying machine learning to the task of TI-RADS classification, the following considerations should be made:Recognize and integrate the presence of cystic and/or solid components within the nodule to recompute its composition;Adapt the echogenicity model to yield regression outputs, facilitating the measurement of the nodule’s contrast;An ill-defined margin may be disregarded, as the model possesses the ability to discern the nodule’s border with greater precision than the human eye.

Analysis of the models with Grad-CAM revealed several insights. Informed by these Grad-CAM insights, we modified the output of the neural network. In the initial version of the model, the output layer for Composition was a four-unit layer with softmax, corresponding to cystic composition, solid composition, mixed composition, or spongiform. We replaced the four outputs with three independent logistic sigmoid outputs corresponding to cystic, solid, and spongiform, programmatically outputting mixed when both cystic and solid outputs were above a threshold of 0.5. The echogenicity output layer, which originally had four softmax units, was replaced with a single linear output with target values zero, one, two, and three, coding the classes hyperechoic, hypoechoic, very hypoechoic, and anechoic. Inference was counted as correct if the output’s nearest integer was the target class. Additionally, we removed the margin output unit corresponding to the ill-defined shape class and consolidated it with the “smooth” shape margin. This can be justified clinically, as the TI-RADS scores for these four margin types are identical. This streamlined model, post-training, displayed enhanced results, as presented in Table 3.

The results of this investigation furnish robust evidence affirming the coherence between automated classification in alignment with ACR TI-RADS guidelines and prevailing radiology theory. This consonance suggests that AI could prove an instrumental asset for physicians and radiologists in the precise detection and classification of ACR TI-RADS and malignancies.

## 5. Discussion

This study sheds light on the capabilities and limitations of AI applications in the realm of radiology, emphasizing alignment with analytical approaches employed by human radiologists. The investigation underscores the potential of AI systems to augment radiological practice, though some challenges persist, particularly in differentiating between micro-calcifications and macro-calcifications based on the size of calcification.

An important finding of this research is the effectiveness of AI systems in the analysis of medical images for the identification and classification of suspicious nodules. This capability has potential clinical significance for early detection and diagnostic accuracy concerning potential malignancies.

Automated TI-RADS assessment presents several benefits. It enhances the rapidity and precision of image data analysis, potentially mitigating human errors and augmenting overall efficiency. The continuous learning attribute of ML models, predicated on extensive datasets, allows incremental improvements in accuracy and performance, providing a dynamic tool for radiologists.

However, it is essential to position the automated system within the broader clinical context, acknowledging it as a complementary asset rather than a standalone solution. The expertise and nuanced judgment of human radiologists remain indispensable in the diagnostic process. We envision the system as a collaborative tool that refines radiologists’ skills, enabling them to concentrate on intricate cases and augmenting their confidence in the assessments.

We note that human radiologists can benefit from a symbiotic relationship with intelligent systems for thyroid nodule diagnosis. By capitalizing on the distinct advantages of both entities, we propose a more sophisticated and accurate evaluative approach. The integration of AI in this manner with the “human in the loop” has the potential to drive thoughtful evolution in medical practice, both maintaining the essential human element and embracing technological innovation to enhance care.

While the experiments presented here provide insights into the integration of AI and traditional radiology practices, our work has some constraints:Model Selection: Our study focuses primarily on two deep learning architectures, ResNet50 and DenseNet201. The inclusion of additional models might have provided a more comprehensive view of the capabilities across different architectures;Dataset Constraints: The study’s findings are based on a specific dataset, which may limit the generalizability of the results to other contexts or types of nodules;Differentiation Challenges: As noted, challenges in differentiating between micro-calcifications and macro-calcifications remain a complex issue that warrants further investigation;Unexplored Factors: There may be other clinical or technical factors that were not explored in this study that could influence the results.

These limitations delineate potential areas for future research and refinement, providing a roadmap for further exploration and development in this field. By transparently acknowledging these constraints, we hope to foster a more nuanced and critical understanding of our study’s contributions and implications.

## 6. Conclusions

This study explores integration of machine learning and the clinician-oriented ACR TI-RADS framework. We advocate further work to utilize the technology in clinical practice.

Our findings reveal the capability of ML systems to analyze vast volumes of image data with precision, reducing the margin for human error and contributing to the overall efficiency of the diagnostic process. However, we emphasize again that the system’s role is and should be complementary, offering an additional layer of support to radiologists rather than replacing their experiential knowledge and clinical judgment.

This research indicates the potential for more robust, nuanced, and potentially more accurate evaluations in thyroid nodule diagnosis. The weaknesses of the current model, such as its difficulty in distinguishing between micro-calcifications and macro-calcifications, indicates a need for future exploration and development.

In conclusion, we advocate integration of AI within the medical field, particularly in radiology. Machine learning algorithms can complement human expertise, leveraging the strengths of both to enhance healthcare practices and patient outcomes. The study not only shows how machine learning can be useful in medical imaging but also offers a blueprint for construction of new tools that can continually adapt, learn, and evolve to meet the dynamic needs of modern healthcare.

## Figures and Tables

**Figure 1 sensors-23-07289-f001:**
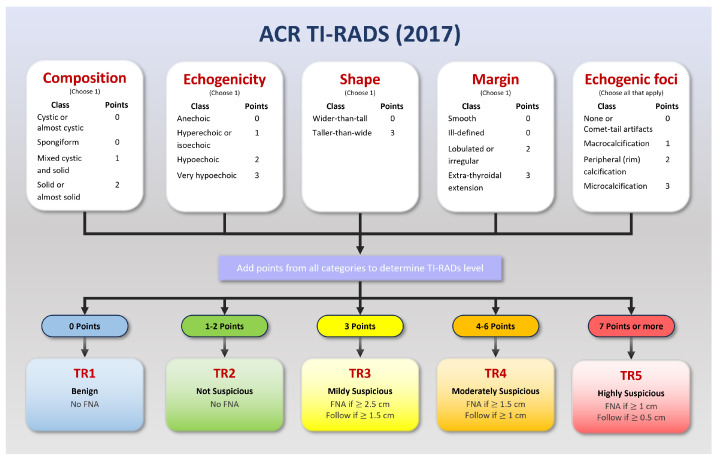
ACR TI-RADS 2017: A risk stratification system for thyroid nodules derived from ultrasound findings. This system assigns scores ranging from 1 to 5, where higher scores correspond to an increased probability of malignancy. It plays an essential role in guiding the decision-making process for further evaluation and management of thyroid nodules. Adapted from Tessler [13].

**Figure 2 sensors-23-07289-f002:**
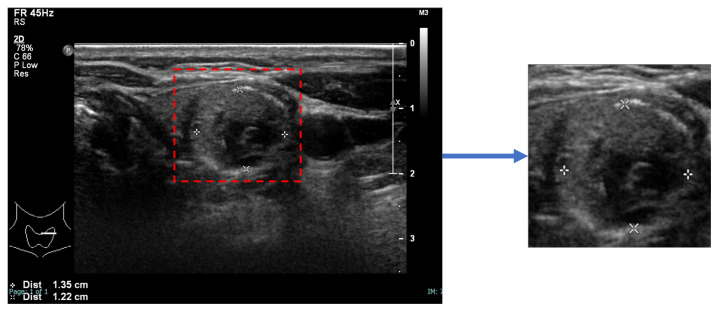
Preparation of a nodule image. The nodule, segmented using StableSeg GANs, was cropped with a 5% margin and then resized to 512 × 512 pixels.

**Figure 3 sensors-23-07289-f003:**
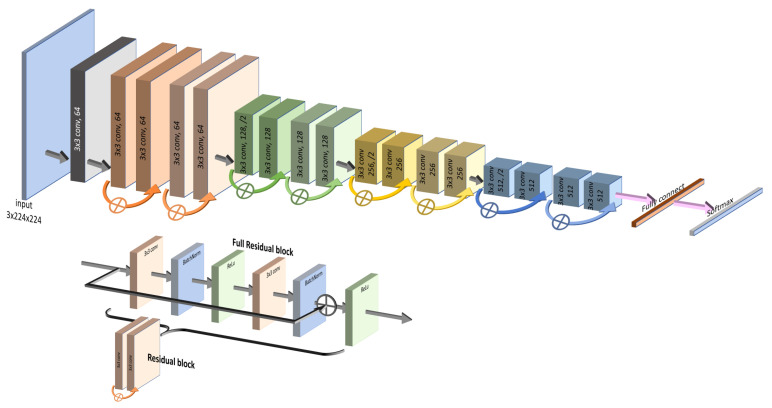
ResNet structure (**top image**). The ResNet model contains multiple residual blocks (**bottom image**), each with skip connections that allow the input to bypass one or more layers, focusing on more recent features.

**Figure 4 sensors-23-07289-f004:**
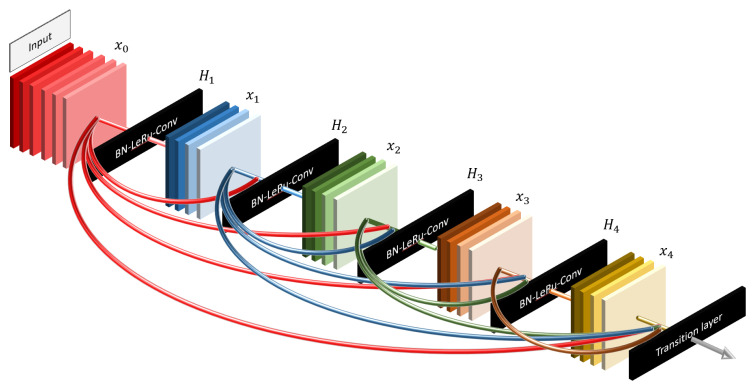
DenseNet structure. DenseNet promotes dense connectivity between layers, allowing for efficient feature reuse and improved gradient flow, ultimately enhancing model performance. Adapted from Huang et al. [20].

**Figure 5 sensors-23-07289-f005:**
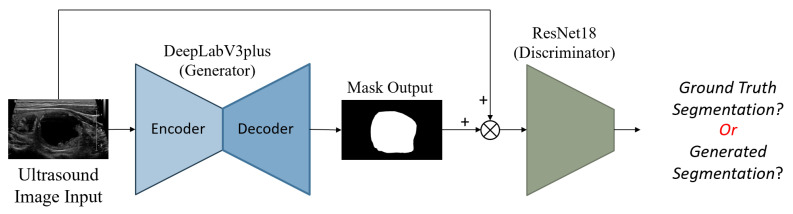
StableSeg GANs. This GANs-based segmentation model utilizes DeepLabV3+ as a generator and ResNet16 as a discriminator. Reprinted from Kunapinun et al. [24].

**Figure 6 sensors-23-07289-f006:**
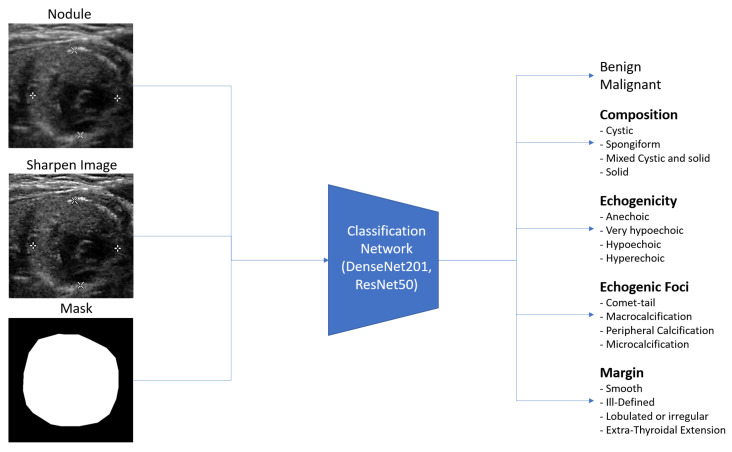
Overall input and output of the model, illustrating the integration of different image tensors and the multiple classification outputs for TI-RADS.

**Figure 7 sensors-23-07289-f007:**
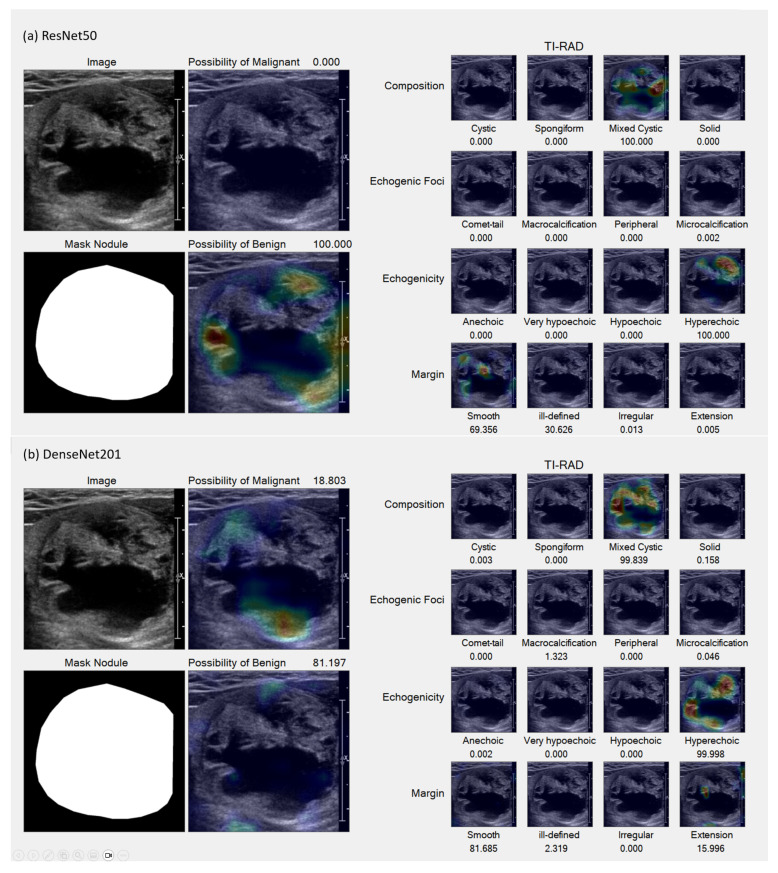
Samples of accurate results. (**a**,**b**) Benign lesion exhibiting well-defined mixed solid–cystic composition, classified under TI-RADS-2, thereby denoting a low suspicion for malignancy (1.5%). Both ResNet50 (**a**) and DenseNet201 (**b**) models accurately identified mixed solid–cystic components, the hyperechoic solid region, and the smooth border, all of which contributed to a high likelihood of a benign diagnosis.

**Figure 8 sensors-23-07289-f008:**
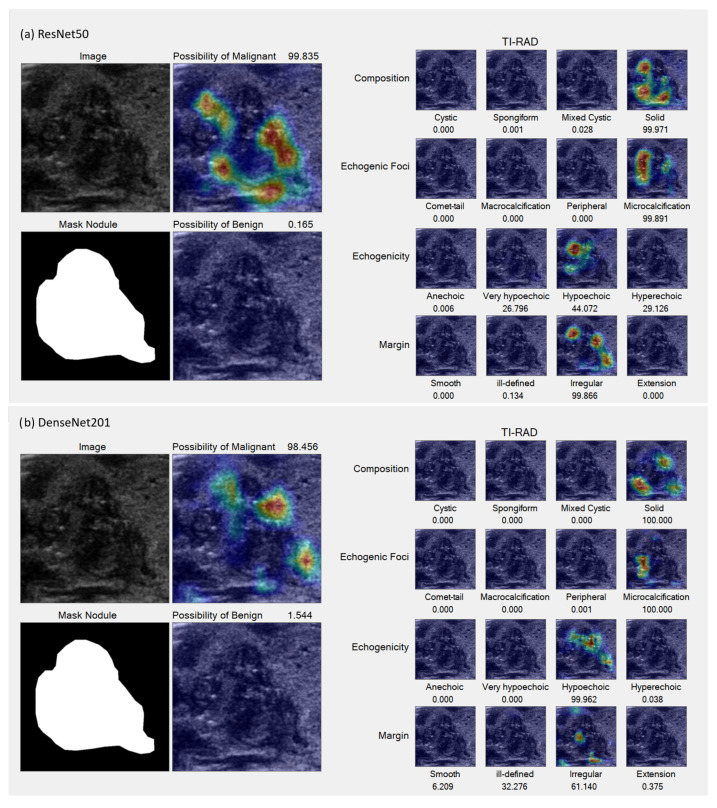
Sample accurate results. (**a**,**b**) TI-RADS-5 malignant nodule characterized by hypoechoic composition, lobulated margins, internal micro-calcification, and a taller-than-wide appearance, all indicative of a high suspicion for malignancy (35%), which was subsequently confirmed pathologically. Both the ResNet50 (**a**) and DenseNet201 (**b**) models accurately identified the regions of micro-calcification and irregular borders. However, slight variations can be observed in the echogenicity and margin details between the two models.

**Figure 9 sensors-23-07289-f009:**
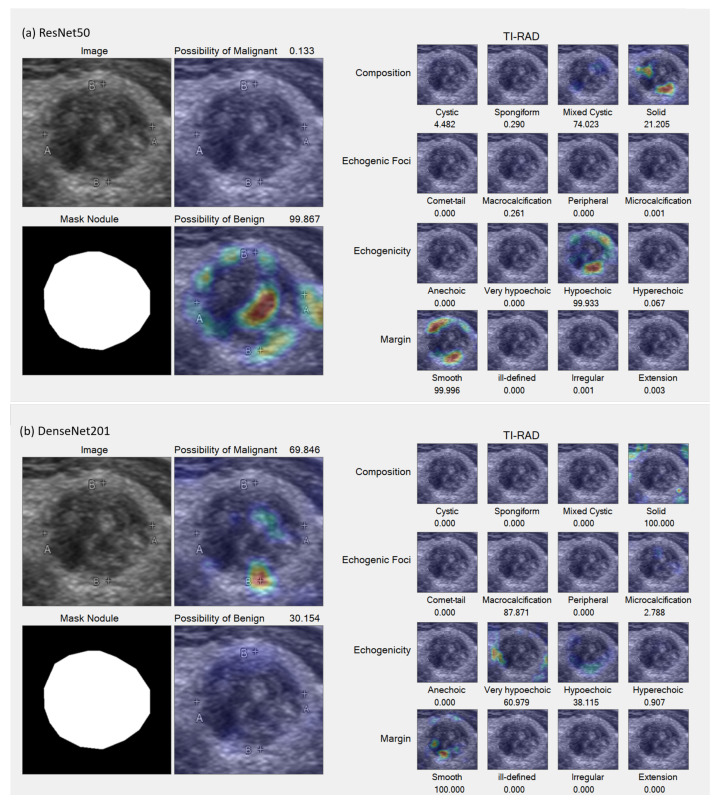
Sample incorrect result. (**a**,**b**) Malignant lesion characterized by a well-defined iso to very hypoechoic solid composition and internal micro-calcifications, classified as TI-RADS-5, indicating a high suspicion for malignancy (35%). The ResNet50 model (**a**) incorrectly suggested a higher likelihood of benignity, failing to detect the very hypoechoic component, in contrast to the DenseNet201 model (**b**), which indicated a greater possibility of malignancy by successfully identifying this component. Nevertheless, both models were unable to detect the internal punctate echogenic foci.

**Table 1 sensors-23-07289-t001:** Summary of the data used for training and testing, including the number of benign and malignant nodules.

Dataset	Benign	Malignant
Training	603	725
Testing	70	79
Total	673	804

**Table 2 sensors-23-07289-t002:** Accuracy of Nodule Malignancy Classification and TI-RADS Classification.

Classification Type	ResNet50 (%)	DenseNet201 (%)
Benign vs. Malignant	74.038	62.500
Composition	87.500	83.654
Echogenicity	62.500	66.346
Margin	54.808	53.846
Comet-Tail	99.835	99.835
Macro-Calcification	87.500	84.615
Peripheral-Calcification	99.077	97.115
Micro-Calcification	95.677	76.923

**Table 3 sensors-23-07289-t003:** Analysis of Malignancy Nodule and TI-RADS Classification.

Classification Type	ResNet50 (%)	DenseNet201 (%)
Benign vs. Malignant	74.763	62.687
Composition	89.423	84.615
Echogenicity	69.231	72.115
Margin	77.885	78.641
Comet-Tail	100.000	100.000
Macro-calcification	86.765	84.959
Peripheral-Calcification	98.781	97.000
Micro-calcification	95.865	76.264

## Data Availability

Code available at https://github.com/Alisa-Kunapinun/tirads_classification.git (accessed on 8 August 2023).

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
