# Peer review of "Explainable Automated TI-RADS Evaluation of Thyroid Nodules"

_sensors, 2023, doi:10.3390/s23167289_

Round 1

Reviewer 1 Report

 This manuscript employs ResNet and DenseNet for classification of the nodule types and compares the results with that of the Ti-RADs. This manuscript aims to investigate whether there is alignment between AI classification in accordance with the ACR TI-RADs guidelines and established radiology theory.

1.  The motivation of this study is ambiguous. The authors should present it clearly in the Introduction section.

2.  The authors stated in the main text (Page 5, line 164) that there are three images as the input (an ultrasound image, a sharpened version of the ultrasound image, and a segmented nodule image), whereas there are only two input images in Fig. 6.

3.  Using TI-RAD outputs as the golden standard to evaluate the performance of the classification network is doubtful. The results presented in this manuscript can only demonstrate that existing advanced alignment neural networks’ performance has a congruence with TI-RAD to some extent. The 60-70% accuracy (Table I) cannot show a strong congruency. Whether the results can serve as a strong support in doctors’ decision making is doubtful.

4.  Page 2, line 51, the reference is missing.

English needs polishing by a native speaker.

Reviewer 2 Report

1. Some authors did not provide their affiliations. 

2. Some quantitative results should be presented in abstract.

3. Reference in Line 52 is missing.

4. Please standardized the usage of Grad-Cam and Grad-CAM within the text. I believe the latter one is more accurate.

5. Problem statement and research challenges that motivate the current study are not clearly explained. Please elaborate to better clarify the significance of current research.

6. The novelty and technical contributions of current work need to be better explained and elaborated.

7. Sentences in Lines 76 and 77 can be merged with those in Lines 70 to 75. It is unnecessary to have a new paragraph with one sentence only.

8. Quality of Figure 1 is poor and need further enhancement. 

9. Related works covered in Sections 2.4 and 2.5 are insufficient. A more extensive literature review related to Deep Learning for Medical Imaging and Grad-CAM for Medical Imaging need to be covered.

10. Furthermore, the authors need to explain the main differences between their proposed works and those existing works. 

11. The data characteristic of ultrasound images used for training, validation and testing datasets should be summarized in table form. 

12. Further explanation is needed to justify the importance of Eg. (2). What is the meaning of image_gradcam and why it appears at both side of equation? It is also not clear how this equation can be useful to explain the classification results made by deep learning to expert radiologists. 

13. It is not clear why the authors only limited their studies on both ResNet50 and DenseNet201. It would be more interesting to see if the authors can expand their studies on other pretrained networks and investigates if the same findings can be observed through Grad-CAM. 

14. A more comprehensive conclusion needs to be presented. 

15. The limitation of current works should be discussed. 

16. Please highlight some future works that can be extended from current study.

17. Authors are encouraged to share their source codes so that other researchers in similar areas can be benefited. 

Some grammatical errors and typos are found in manuscript. Authors need to improve the quality of English language before resubmission.

Round 2

Reviewer 1 Report

The authors made a considerable revision of this manuscript. All my concerns have been addressed properly. I recommend acceptance of this manuscript.